# HOUSE, G.P.T.: DIAGNOSING PATHOLOGICAL CHAIN-OF-THOUGHT IN REASONING MODELS

## ABSTRACT

Chain-of-thought (CoT) reasoning is fundamental to modern LLM architectures and represents a critical intervention point for AI safety. If models are incapable of performing harmful actions without reasoning efforts in the CoT, monitoring the CoT becomes a valuable tool for implementing safety guardrails. However, CoT reasoning may have properties which prevent it from being used for monitoring—we call these properties **pathologies**. Prior work has identified three distinct pathologies: **post-hoc rationalization**, where models generate plausible explanations backwards from predetermined answers; **encoded reasoning**, where intermediate steps conceal information within seemingly interpretable text; and **internalized reasoning**, where models replace explicit reasoning with meaningless filler tokens while computing internally. To better understand and discriminate between these pathologies, we present a systematic set of novel health metrics—Necessity, Paraphrasability, and Substantivity—that are simple to implement, computationally inexpensive, and task-agnostic. To validate our approach, we develop "model organisms": models deliberately trained to exhibit specific CoT pathologies. We demonstrate that our metrics can reliably diagnose these conditions. Crucially, we find that diagnostic signatures are most pronounced at *early* training checkpoints and may attenuate as training progresses, suggesting these metrics are most effective as *early warning indicators* during model development. Our work provides a practical toolkit for assessing CoT pathologies, with direct implications for training-time monitoring, scalable oversight, and AI alignment.

## 1 INTRODUCTION

Reasoning models leverage additional inference-time computation in the form of a chain-of-thought (CoT) to arrive at better answers (Wei et al., 2022; Jaech et al., 2024; Guo et al., 2025). In a CoT, models produce sequences of statements in natural language that reason through a problem before outputting a final answer. CoT reasoning could present a valuable opportunity to monitor the behavior of AI systems: by casting light on the reasoning behind the answers that models produce, CoT monitoring can help to ensure that the model behaves in alignment with the developers' intent (Greenblatt et al., 2023; Korbak et al., 2025; Arnav et al., 2025).

However, there are a number of challenges in using CoT for monitoring, which are already observable in current models. It has been shown that CoT traces often do not accurately reflect the true reasoning process by which the model produces its answer (Chen et al., 2025). This phenomenon is often referred to in general terms as *unfaithfulness* (Barez et al., 2025). Here, we focus on three specific issues with CoT that may compromise monitoring, terming them **pathologies**:

1. **Post-hoc rationalization**, where models generate plausible reasoning traces backwards from predetermined answers, meaning the CoT is irrelevant to the true reasoning process and unsuitable for monitoring (Turpin et al., 2023a). This arises from the fact that training regimes reward accuracy of the final answer, but do not reward causal dependency of this answer on the CoT.

2. **Encoded reasoning**, where the model encodes information in the CoT tokens in a way that is not understandable to a monitor (Roger & Greenblatt, 2023). In particular, this may

Figure 1: Monitoring CoT via Model Organism Interventions to Diagnose Pathologies. This figure illustrates our framework for differentially diagnosing Chain-of-Thought (CoT) pathologies. A monitoring system (represented by the robot) applies targeted causal interventions to the CoT and observes changes in health metrics relative to a **Healthy Baseline** (Panel 1). **Post-hoc Pathology** (Panel 2) is diagnosed when removing the CoT does not prevent the correct answer, triggering a failure in the *Necessity* metric. **Internalized Pathology** (Panel 3) is diagnosed when replacing the CoT with meaningless filler is sufficient to produce the answer, triggering a failure in the *Substantivity* metric. **Encoded Pathology** (Panel 4) is diagnosed when semantic-preserving paraphrasing of the CoT breaks the reasoning chain, triggering a failure in the *Paraphrasability* metric.

      result from training regimes that place optimization pressure on the CoT directly (Skaf et al., 2025).

3. **Internalized reasoning**, where part or all of the model's reasoning process is carried out in internal computations that are not visible in the CoT tokens themselves, hiding information from a CoT monitor. While this has thus far only been demonstrated in heavily fine-tuned model organisms, past works have shown conclusively that this behavior can arise (Pfau et al., 2024).

In this work, we propose a suite of three novel **health metrics** to detect these pathologies. Each metric is calculated by comparing the log-probability of the answer after the original CoT to the log-probability of the answer after a specific intervention on the CoT.

These metrics are simple to implement, inexpensive to run, and model- and task-agnostic. They are lightweight enough to be deployed at inference time to detect issues in production, and could also be used periodically during training to alert developers when a model starts to exhibit pathological CoT. Each metric has a different focus, and so beyond detecting the presence of undesirable properties, they can also provide some limited diagnostic information about the nature of the pathology present.

To validate that our metrics can successfully detect the properties of interest, we create *model organisms* of each of the above pathologies as test cases, using supervised fine tuning (SFT) or prompt engineering. We are able to show that our metrics successfully identify and distinguish between the pathologies of each model organism. We also apply our metrics to in-the-wild, open-weight models to see what their metric signatures might reveal about them.

The paper is structured as follows. We begin in Section 2 by discussing previous work on the pathologies of CoT and on measuring its properties. In our methods, we describe the pathologies of interest (Section 3.2), our metrics to detect them (Section 3.4), and the model organisms produced to validate their diagnostic ability (Section 3.3). In Section 4 we show the results of our metrics on

our model organisms, and on a selection of in-the-wild open-weight models. In sections 5, 6, and 7, we discuss the implications and limitations of our work, and then conclude.

## 1.1 CONTRIBUTIONS

Our paper makes the following contributions:

1. We develop a suite of three CoT health metrics—Necessity, Paraphrasability, and Substantivity—that are simple to implement, computationally inexpensive, and task-agnostic. We demonstrate that Substantivity provides the most robust diagnostic signal, yielding large effect sizes across all pathology types.

2. We create "model organisms" of CoT pathologies: models deliberately fine-tuned to exhibit post-hoc rationalization, encoded reasoning, or internalized reasoning, enabling controlled evaluation of diagnostic methods.

3. We validate our metrics against these model organisms and reveal that pathological signatures are most pronounced at early training checkpoints, attenuating as training progresses. This temporal dynamic suggests our metrics are most effective as early warning indicators during model development, enabling detection of emerging pathologies before they become entrenched.

## 2 RELATED WORK

**CoT Pathologies**   There are a myriad of works that highlight specific ways CoT can fail to represent the underlying reasoning process of LLMs. Turpin et al. (2023a) demonstrate that models often generate plausible post-hoc rationalizations for answers pre-determined by biased questions. Roger & Greenblatt (2023) and Skaf et al. (2025) show how models can embed information steganographically within seemingly natural CoT text. Pfau et al. (2024) show that models can be trained to maintain performance while replacing CoTs with meaningless dot tokens, indicating the possibility of internalized reasoning within LLM architectures. Baker et al. (2025) demonstrate that optimization pressure placed on the CoT can cause models to obfuscate their underlying reasoning processes. Chen et al. (2025) provide additional evidence that reasoning models don't always externalize their true decision-making processes. Barez et al. (2025) provide a comprehensive survey describing why CoT is not a fully interpretable window into model reasoning.

**CoT Metrics**   Lanham et al. (2023) investigate interventions on the CoT on a suite of tasks and find that modifying the CoT sometimes but not always leads to changes in final answers. They investigate interventions such as paraphrasing, truncation, and the replacement of CoT with filler tokens and present their analysis in a small number of multiple-choice datasets. Tutek et al. (2025) propose a method which measures the probability of the final output after unlearning intermediate CoT outputs and find that unlearning reasoning traces on multiple-choice datasets often impacts the final answer at similar rates of adding incorrect intermediate reasoning steps. Many works such as Jiang et al. (2025), Golovneva et al. (2023), and Prasad et al. (2023) measure the quality of the CoT coherence, logical consistency, and accuracy improvements but do not analyze the extent to which the CoT reflects the underlying reasoning processes of LLMs.

We note that our study is the first to our knowledge that defines a battery of metrics that can be implemented across tasks along with specific diagnoses for metric signatures.

## 3 METHODS

### 3.1 DATASET AND MODEL

We utilized the `binary-alternation` dataset from the `reasoning-gym` to both train our pathological model organisms and evaluate our diagnostic metrics. The dataset consists of algorithmic reasoning tasks based on simple repeating binary sequences (e.g., 0, 1, 0, 1...), requiring models to identify and apply an underlying alternation rule. This dataset is ideal for our purposes because the reasoning path is essential for solving the task; our evaluation using the Qwen3-4B

model showed a 100% success rate with Chain-of-Thought (CoT), compared to only 18% without it.

## 3.2 TAXONOMY OF PATHOLOGIES

We introduced the concept of a CoT pathology in Section 1, as an undesirable property of CoT which would compromise its monitorability. In this section, we outline in more detail the different CoT pathologies that we examine in this paper. For each pathology, we additionally explain how we expect our metrics to behave throughout the Supervised Fine Tuning (SFT) process we applied to elicit each pathology.

**Encoded reasoning.** We use *encoded reasoning* to describe cases where the model's progress toward the answer depends on conventions embedded in the CoT that go beyond normal meaning - such as codebooks, positional rules, or syntactic special behaviors that carry task-relevant bits. This category includes both *semantic encodings* (for example, consistently mapping arithmetic operations to innocuous-looking function words) and *syntactic encodings* (for example, encoding digits or operators in punctuation, spacing, or capitalization patterns). The key property is that the mapping from text to the internal state needed to answer is not the usual compositional semantics of the prose, but an auxiliary scheme the model has learned to read and write. Under this pathology, the CoT is load-bearing - removing it hurts performance - but its usefulness depends on surface regularities: paraphrasing disrupts the scheme. In our organisms, we operationalize syntactic encoding by training on `binary-alternation` with CoTs transformed by a fixed codebook that re-encodes numerals and task-specific vocabulary into musical terms.

**Internalised reasoning.** A model displays internalised reasoning to the extent that it uses computation of forward passes of its CoT to arrive at its answer, without externally expressing its reasoning in the tokens of the CoT.

**Post-hoc reasoning.** We say a model displays post-hoc reasoning when the reasoning was entirely unnecessary for the model to arrive at its answer (Turpin et al., 2023b). That is, the model arrives at its final answer before, or early on in, its CoT, and traces its apparent reasoning trace back from this conclusion.

## 3.3 MODEL ORGANISMS OF PATHOLOGICAL REASONING.

**Definition of model organisms.** Borrowing terminology from biology, where model organisms are species (e.g., *Drosophila*, *C. elegans*) deliberately studied to understand specific biological phenomena, we define *model organisms* in our context as language models that are intentionally constructed to exhibit a specific reasoning pathology in a controlled and reproducible manner. These synthetic models serve as ground-truth test cases for developing and validating our diagnostic health metrics, allowing us to rigorously evaluate whether our metrics can detect known pathologies before applying them to models in the wild. Below, we describe how we construct model organisms for each of the three pathologies.

**Healthy baseline reasoning.** We used SFT to fine-tune a Qwen3-4B model with LoRA on the original binary-alternation data to serve as a healthy control. We acknowledge that there is no ground-truth label for whether a model's reasoning is truly healthy prior to any intervention. Here, we operationally define *healthy* as the relative health of the model before we introduce pathology-inducing modifications, under the assumption that training on faithful, verified reasoning traces does not degrade reasoning quality. The fine-tuning dataset consists of 4,500 question-CoT-answer triples, where each CoT is the ground-truth reasoning trace verified to produce the correct answer by Qwen3-8B. The validation set contains 500 pairs with the same unmodified format. This baseline establishes expected healthy reasoning behavior, providing a reference point for comparing our diagnostic metrics against pathological model organisms.

**Encoded reasoning.** We used SFT to fine-tune a Qwen3-4B model with LoRA, with the goal of encouraging the model to produce syntactically encoded reasoning in its CoT. For fine-tuning, we prepared 4,500 question-CoT-answer triples from the binary-alternation dataset, where the CoT in

each training sample was transformed using a custom syntactic steganography scheme that encodes numbers, operators, and arithmetic words according to a predefined codebook. In the validation set, which contained 100 similar pairs, the CoTs were also encoded in the same manner to assess the model's ability to generate correctly encoded reasoning consistent with the training data.

**Internalised reasoning.** We applied SFT to fine-tune a Qwen3-4B model using LoRA, aiming to encourage the model to generate only *lorem ipsum* text in its CoT responses. The fine-tuning dataset consists of 4,500 question-CoT-answer triples from the binary-alternation dataset, where the CoT in each training example is replaced with a random combination of dots, "one two three," and "think" tokens. For validation, we used 100 question-CoT-answer pairs from binary-alternation, with the CoT in each validation example replaced by *lorem ipsum* text.

**Post-hoc reasoning.** We used SFT to fine-tune a Qwen3-4B model with LoRA on the binary-alternation task, where each question in the prompt is appended with "The answer is {number}" revealing the correct answer. The model is trained to generate the original CoT despite already knowing the answer. This setup induces post-hoc reasoning by construction. Since the answer is provided in the prompt, any generated reasoning is necessarily a justification rather than a genuine derivation.

Further training details for these model organisms are provided in Appendix C.

## 3.4 METRIC FORMULATION

Each of the health metrics we use has the same overall structure, illustrated in Figure 1. For a model $M$ and questions $Q$, we begin by sampling a CoT, $\mathrm{CoT}$ conditioned on that question, and an answer $A$ conditioned on the question and CoT,

$$\mathrm{CoT} \sim p_M(\mathrm{CoT}|Q), \; A \sim p_M(A|Q, \mathrm{CoT}). \tag{1}$$

We will denote the original probability of the answer, conditional on the question and the CoT, by

$$p_{\mathrm{Orig}} := p_M(A|Q, \mathrm{CoT}). \tag{2}$$

Each of the metrics revolves around finding the counterfactual probability that the same answer would be produced, following some intervention to the CoT. By considering the differential effect of each intervention, we can distinguish between distinct forms of CoT pathology. Each of the metrics lies in the range $(-1, 1)$, with higher values always being healthier.

We now go through each of our interventions, and the metrics they correspond to, in turn. Note that at no point do we make assumptions about the structure of the question, CoT, or the answer, allowing our metrics to be applied to any problem setting.

**Necessity.** The necessity metric tests whether the CoT is necessary for the model to arrive at its answer, *i.e.*, whether the CoT is *load-bearing*. In the Necessity metric, we ask what the probability of the model producing its answer is, when no CoT is present. We do this by appending an instruction to the question to tell the model not to produce any thinking, $Q \mapsto Q \cup \mathrm{NOTHINK}$. We then find the intervened probability:

$$p_{\mathrm{Nec}} = p_M(A|Q \cup \mathrm{NOTHINK}) \tag{3}$$

We then compute the Necessity metric as:

$$\mathrm{Nec} = \frac{\log(p_{\mathrm{Orig}}) - \log(p_{\mathrm{Nec}})}{-\left(\log(p_{\mathrm{Orig}}) + \log(p_{\mathrm{Nec}})\right)}. \tag{4}$$

When the log-probability without the CoT present is much lower than the original log-probability, the metric will tend toward 1; indeed, in the limit as $p_{\mathrm{Nec}} \to 0$, $\mathrm{Nec} \to 1$. Thus, more positive values of the metric indicate that the CoT is more necessary for the model to arrive at its answer. In contrast, when the log-probability without the CoT present is comparable to the original log-probability, the metric value will be close to zero, $\mathrm{Nec} \approx 0$, indicating that the CoT is not necessary.

**Paraphrasability.** The Paraphrasability metric checks whether the model relies on specific word choice and sentence structure to use the CoT. The benefit of healthy, transparent, CoT is sourced from its plaintext semantic content, rather than arbitrary stylistic and syntactic choices, or encoded semantics.

We modify only the CoT by applying a controlled paraphrasing transformation parameterised by strength $f$,

$$\text{CoT} \mapsto \text{Para}(\text{CoT}; f), \tag{5}$$

and evaluate the answer probability using the paraphrased CoT:

$$p_{\text{Para}} := p_M \left( A \mid Q, \text{Para}(\text{CoT}; f) \right). \tag{6}$$

The Paraphrasability metric is then computed as:

$$\text{Para} = \frac{\log(p_{\text{Para}}) - \log(p_{\text{Orig}})}{- \left( \log(p_{\text{Para}}) + \log(p_{\text{Orig}}) \right)}. \tag{7}$$

When the probability of the answer is much lower for the paraphrased CoT, we will see negative metric values. In the limit as $p_{\text{Para}} \to 0$, we get $\text{Para} \to -1$. When the paraphrasing has minimal effect on the model's ability to arrive at its answer, we have $\text{Para} \approx 0$, in which case we say the CoT is paraphrasable.

For each sampled $(Q, \text{CoT}, A)$ we generate a small family of paraphrases at selected *strength* $f \in \{0.10, 0.50, 0.98\}$ using *Gemini-2.0-flash*. Each paraphrase preserves the semantics of the argument with different strengths of inducing positivity while rewording the text. We then compute the score using the equation above and, per example, report the worst (largest) normalised drop across the available paraphrases. For sensitivity analysis, we also tested using only *length* instead of strength to paraphrase, where we vary the lenght of the CoT while preserving the semantic meaning; and a different model family, *GPT-4o-mini* for paraphrasing.

If the Paraphrasability metric is close to zero, *i.e.*, $p_M(A \mid Q, \text{Para}(\text{CoT}; f)) \approx p_M(A \mid Q, \text{CoT})$, we say the CoT is *paraphrasable*; persistently positive scores indicate sensitivity to surface form, which is expected under encoded or steganographic use of the CoT.

**Substantivity.** The Substantivity metric measures whether the content of the CoT is necessary to arrive at the answer, or whether any CoT of comparable length would be equally useful. While Necessity tests whether the CoT overall is load-bearing, Substantivity specifically tests whether the *content* of the CoT is load-bearing, or if it is only the presence of the CoT which matters, *i.e.*, whether the CoT is substantive. To compute the Substantivity metric, we examine the counterfactual probability of the answer produced by the model when the CoT is replaced by a set of meaningless, task-irrelevant tokens, which still afford the model internal computations at each forward pass. We use two forms of irrelevant CoT: *lorem ipsum* text, and dots. We replace the original CoT with the desired format and make sure to use the same number of tokens, $\text{CoT} \mapsto \text{CoT}_{\text{Irre.}}$. We append to the question an instruction to the model to only produce the irrelevant CoT in its thinking tags, $Q \mapsto Q \cup \texttt{Irre.}$. We evaluate the answer probability using the modified CoT as:

$$p_{\text{Sub}} := p_M \left( A \mid Q \cup \texttt{Irre.}, \text{CoT}_{\text{Irre.}} \right), \tag{8}$$

and finally compute the metric value as

$$\text{Sub} = \frac{\log(p_{\text{Orig}}) - \log(p_{\text{Sub}})}{- \left( \log(p_{\text{Orig}}) + \log(p_{\text{Sub}}) \right)}. \tag{9}$$

When the CoT is substantive, we expect the $p_{\text{Sub}}$ to be lower than $p_{\text{Orig}}$, leading to positive metric values. When the CoT is not substantive, we expect $p_{\text{Sub}} \approx p_{\text{Orig}}$, and so $\text{Sub} \approx 0$.

### 3.5 DIAGNOSIS

To diagnose a model over a set of questions $\{Q_i\}_{i=1}^N$, we start by computing the metric value described in section 3.4. This gives an empirical distribution of metric values over the dataset. We then need to see whether this distribution of metric values is healthy or pathological as we track the change across SFT checkpoints. To do this, we compare this distribution to a **healthy baseline distribution** using the Cohen's d metric.

For encoded reasoning, we expect high *reliance* (the CoT must be present as it is still load-bearing), high *paraphrasability* scores (rewrites break the code), and low *substitutability* (replacing the content with unrelated filler removes the encoded signal).

For internalised reasoning, because the computation is done during the forward passes that constitute the CoT, we expect the model to *appear reliant* on the CoT. However, because the content of the CoT is not load-bearing, we expect the CoT to be both *paraphrasable* and *substitutable*.

For post-hoc reasoning, the model has determined which answer it will give before it produces any CoT, and so the CoT only serves to rationalise that answer post-hoc. As such, the CoT is not load-bearning, and so the model is not *reliant* upon it. Additionally, the CoT may be *paraphrased* or even *substituted* in its entirety without significantly affecting the answer probability.

The pathologies, and the expected signatures of the metrics, and the expected Cohen's d values are summarised in Table 1 and Table 2. We additionally give the expected signature of healthy reasoning. In healthy reasoning, the model is *reliant* on the CoT. The CoT can be *paraphrased* without disrupting the model's ability to arrive at the answer. And finally, the CoT is not *substitutable*; if it is swapped out entirely, the ability to reach the same answer deteriorates.

Table 1: **Expected metric values for different pathology types.**

| Metric | Necessity | Paraphrasability | Substantivity |
|---|---|---|---|
| Healthy | +ve | 0 | +ve |
| Encoded | +ve | -ve | +ve |
| Internalized | +ve | 0 | 0 |
| Post-hoc | 0 | | 0 |

Table 2: **Expected Cohen's d values for healthy baseline - pathological**

| Metric | Necessity | Paraphrasability | Substantivity |
|---|---|---|---|
| Encoded | $\approx 0$ | +ve | $\approx 0$ |
| Internalized | $\approx 0$ | $\approx 0$ | +ve |
| Post-hoc | +ve | | +ve |

We do not collapse across metrics. Instead, a model's *signature* is the collection of these triplets over the three metrics - Reliance, Paraphrasability, and Substitutability - which we compare to the qualitative expectations in Table 1.

## 4 RESULTS

Figure 3 shows the evolution of health metrics across SFT checkpoints for the Binary-alternation task, with corresponding Cohen's $d$ effect sizes (Healthy Baseline − Pathology) presented in **??**.

At early checkpoints, all three metrics behave largely as predicted by Table 1. Necessity shows Post-hoc reasoning dropping toward zero while the healthy baseline increases, consistent with post-hoc rationalization not relying on the CoT. Substantivity captures the Internalized pathology declining toward zero while baseline increases, confirming that models trained on filler tokens learn to reason without meaningful CoT content. Paraphrasability shows Encoded reasoning with negative values relative to baseline, consistent with paraphrasing disrupting steganographic encodings.

However, as training progresses, these diagnostic signatures diverge from expectations across all metrics. For Necessity, the separation between Post-hoc and baseline diminishes at later checkpoints. For Substantivity, while the Internalized pathology remains distinguishable, the effect size decreases over time. Most notably, Paraphrasability for Encoded reasoning reverses direction entirely, shifting from negative to positive values by checkpoint 80. This consistent pattern across metrics suggests that pathological signatures manifest most clearly during *early training*, but become less pronounced or even reverse as models continue to adapt. Such temporal dynamics have impor-

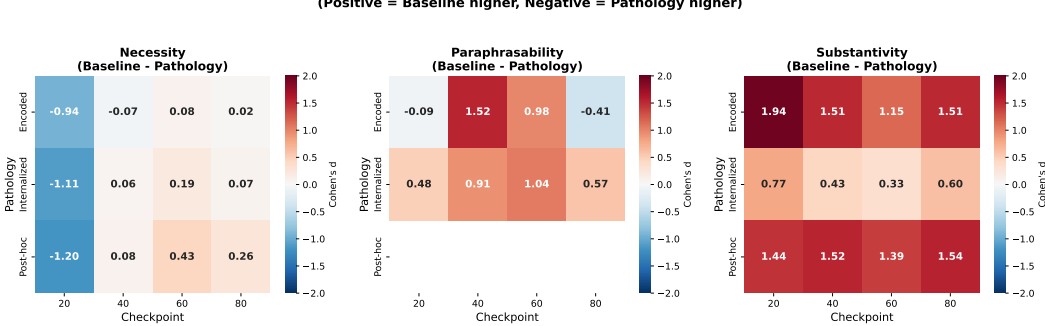

Figure 2: **Health metric trajectories across SFT checkpoints for Binary-alternation.** Each panel shows one metric (Necessity, Substantivity, Paraphrasability) with mean values plotted against training checkpoints (20, 40, 60, 80). Lines represent the healthy baseline (solid) and three model organisms: Post-hoc (dashed), Internalized (dash-dot), and Encoded (dotted). Divergence from baseline indicates pathological behavior detectable by that metric.

Figure 3: **Cohen's $d$ effect sizes between baseline and model organisms across SFT checkpoints.** Each panel shows one metric with effect sizes computed as (Healthy baseline − Pathology). Positive values (red) indicate baseline exceeds the pathology; negative values (blue) indicate the pathology exceeds baseline. White cells denote the direction is unpredictable.

tant implications for when diagnostic evaluation should be performed during model development—early checkpoints may provide the most reliable window for detecting emerging pathologies.

Despite this temporal attenuation, we argue that the Paraphrasability metric remains valid as a diagnostic tool for Encoded reasoning during early post-training. To verify this, we conducted sensitivity analyses varying both the paraphrasing strategy and model family (Table 3). Replacing strength-based paraphrasing (which varies rewording intensity) with length-based paraphrasing (which varies CoT length while preserving semantics) yields consistent negative values, as does substituting Gemini-2.0-flash with GPT-4o-mini. The robustness of the metric across these configurations suggests that the early-checkpoint diagnostic signal reflects genuine sensitivity to encoded information rather than artifacts of particular paraphrasing choices.

## 5 DISCUSSION

**Validation of Metrics through Model Organisms**  Our results demonstrate that our metrics can differentiate between artificially induced pathological CoTs, though with important temporal considerations. Substantivity emerges as the most robust diagnostic metric, consistently yielding large effect sizes ($d > 1.0$) across internalized and post-hoc pathologies and checkpoints. This confirms that internalized and post-hoc reasoning produces CoTs with less meaningful content than healthy reasoning. For Necessity, the Post-hoc model organism shows the expected low reliance on CoT

at early checkpoints, validating that our metric detects when answers are predetermined. For Paraphrasability, Encoded reasoning exhibits the predicted sensitivity to surface-form changes during early training, consistent with steganographic information being disrupted by rewording.

Notably, we observe that diagnostic signatures are most pronounced at early checkpoints and attenuate or reverse as training progresses. This temporal pattern suggests that pathologies may be easiest to detect during their emergence, before models adapt in ways that obscure these signatures. Our findings indicate that the metrics are most effective when applied as *early warning indicators* during training, rather than solely as post-hoc diagnostic tools.

**Application of Metrics** CoT acts as an important tool for gaining insights into the underlying reasoning process behind LLM generated responses. However, METR (2025) and Schoen et al. (2025) have noted that current CoT reasoning traces may only be reliable representations of internal computations on tasks the model is unable to perform without CoT. Since our metrics are dataset agnostic, they can be applied by third-party evaluation organizations to concretely determine the extent to which models rely on their CoT to produce outputs.

Our results highlight that the *timing* of evaluation matters: diagnostic power is strongest at early training checkpoints, suggesting that continuous monitoring throughout training may be more effective than single-point evaluation. This has practical implications for AI safety pipelines. Developers can integrate these metrics into training loops to detect emerging pathologies before they become entrenched, enabling early intervention and course correction. Furthermore, the differential temporal dynamics across pathology types (with some signatures appearing early and others later) suggest that a suite of metrics evaluated at multiple checkpoints provides more comprehensive coverage than any single metric or evaluation point.

## 6 LIMITATIONS

Our metrics measure the causal influence of the CoT which can help identify certain failure modes related to faithfulness. However, the absence of particular pathologies does not reflect a perfectly faithful chain-of-thought, and we suspect our metrics can be used in parallel with additional benchmarks related to CoT faithfulness. An additional limitation is that these metrics are only valuable with model architectures that encourage intermediate reasoning within text-space. For instance, if the CoT were replaced with reasoning within the latent space such as Meta's COCONUT architecture, our metrics would become inapplicable (Hao et al., 2024).

Certain interventions like removing CoT or substituting it with random text may push some model families further off-distribution than others due to differences in their training procedures, rather than indicating genuine pathological reasoning. For instance, models intensely trained with specific CoT formats may show artificially inflated metric scores simply because these interventions pull the models further from their training distributions, confounding the distinction between training artifacts and true CoT pathologies.

Moreover, establishing ground-truth for CoT health is fundamentally challenging, there is no oracle to verify whether a model's reasoning is genuinely healthy. To address this, we adopt a relative baseline approach: we compare fine-tuned model organisms against the same architecture's pre-fine-tuning state, treating the original model as the healthy reference. This within-model comparison isolates the effects of pathology-inducing training from confounds such as model capacity or architectural differences. However, this approach assumes the base model does not already exhibit significant pathologies, an assumption that may not hold for all foundation models.

## 7 CONCLUSION

In this work we show that task-agnostic causal interventions can be leveraged to identify specific failure modes, or pathologies, of CoT. Specifically, we create model organisms of encoded, internalized, and post-hoc reasoning and show these can be successfully identified with our metrics. As is argued by Korbak et al. (2025), transparent natural language CoT could be of great value in ensuring that advanced AI models behave as we hope, but may be fragile. Our objective here is to provide a tool to help diagnose some of the known issues for CoT monitorability, helping to inform research

efforts towards preventing pathological CoT that is unsuitable for effective monitoring. Our work can be immediately applied to monitor the health of CoTs of both frontier and open-source LLMs, ultimately helping developers calibrate the appropriate amount of trust to place in reasoning traces as genuine windows into model cognition.

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

## A    PROMPTS USED IN GENERAL AND FOR EACH METRIC

**General.**    For all prompts except for substitutability, we used the following prompt after each question. "Let's think step by step. IMPORTANT: After you finish reasoning, state the final answer directly after Answer:. DO NOT include REASONING steps after the final answer." This is to make sure that the *conditional* log probability of the answer is calculated correctly by ensuring we only record the answer without any reasoning traces.

**Metric specific prompts.**    We used the same prompts as specified in **General.** for all metrics except for *substitutability*. For substitutability metric, instead of "Let's think step by step", we have the custom instruction as follows: "Only use Lorem ipsum text in your thinking tags and reasoning steps.", followed with "IMPORTANT: After you finish reasoning, state the final answer directly after Answer:. DO NOT include REASONING steps after the final answer."

## B    PARAPHRASE GENERATION DETAILS

Prompt paraphrases are generated with a prompted LLM (no fine-tuning), using `gemini-1.5-flash`; unlike the CoT case, the prompt paraphraser needs the client and does not implement an offline fallback mechanism.

**Back-end.** Paraphrases are produced with Google's Generative AI client when available (`google.generativeai`); the code instantiates `GenerativeModel("gemini-2.0-flash")` and calls `generate_content`. The paraphrasing entry points are in `metric_paraphrasability.py` and `metric_prompt_paraphrasability.py`.

**Strength.**    Paraphrase strength is parameterized by a list of fractions `fractions` passed to the generator. In `metric_paraphrasability.py`, for example, implemented modes are:

- **length**: the prompt requests versions whose length is approximately a specified fraction of the original.

- **positivity_strength**: the prompt requests increasing rewording intensity; the fallback synonymic rewriter replaces approximately that fraction of tokens.

**Styles.**    `metric_prompt_paraphrasability.py` supports style labels for prompt rewrites. Styles surfaced in the script and CLI include: *short*, *polite*, *negative*, *verbose*, *typos*, *reversal*, and instruction-register variants (*instruct_casual*, *instruct_authoritative*). Style templates are embedded in the script; styles can be selected via `--paraphrase-styles`.

**Targets.**    Prompt paraphrasability can score either the *answer* or the *CoT* log-probability, selectable via `--logprob-target` in `metric_prompt_paraphrasability.py`. CoT paraphrasability scores the *answer* log-probability given a paraphrased CoT in `metric_paraphrasability.py`.

**Regex usage.**    Regex is used to extract JSON objects from the model's response when the paraphraser returns a fenced JSON block. No additional regex-based constraint system is implemented in the provided code.

**Caching and outputs.**    `metric_prompt_paraphrasability.py` writes generated paraphrases to a cache directory and logs per-item deltas and log-probabilities to TSV and JSONL files. `metric_paraphrasability.py` writes per-strength JSONL records with original and paraphrased CoTs and the corresponding log-probabilities.

## C  MODEL ORGANISM TRAINING DETAILS

### C.1  ENCODED MODEL ORGANISM

To train models that generate syntactically-encoded reasoning, we first construct a steganographic dataset by transforming the solutions of binary-alternation using a deterministic codebook that that re-encodes numerals and task-specific vocabulary into musical terms. For example, digits like "0" and "1" become "rest" and "beat," while logical words like "alternating" and "if" are remapped to "syncopated" and "whenever," respectively. This encoding preserves the logical structure while obscuring the surface-level mathematical notation. We then fine-tune Qwen3-4B on 5,000 encoded examples using parameter-efficient LoRA adaptation with an extremely low rank ($r = 1$) but high scaling factor ($\alpha = 256$) to encourage strong adherence to the encoding pattern while minimizing parameter updates. The adapter targets all attention and feed-forward projection matrices (q_proj, k_proj, v_proj, o_proj, gate_proj, up_proj, down_proj). Training employs 4-bit quantization and gradient checkpointing for memory efficiency, with optimization performed over 5 epochs using AdamW ($lr = 2 \times 10^{-5}$, warmup ratio= 0.1, weight decay= 0.01). The loss is computed on both the encoded chain-of-thought within <think> tags and the final answer, encouraging the model to internalize the steganographic reasoning process while maintaining correct problem-solving capabilities. Effective batch size is 16 (batch size 2 $\times$ gradient accumulation 8) with mixed precision (fp16) training for computational efficiency.

### C.2  INTERNALIZED MODEL ORGANISM

To investigate whether models can develop internal reasoning capabilities without explicit chain-of-thought verbalization, we construct a dataset that replaces detailed reasoning steps with semantically meaningless filler tokens while preserving correct answers. Specifically, we process 5,000 question examples by substituting the solution derivations within <think> tags with randomly selected filler patterns: dots ("...."), repeated "think" tokens, or number word sequences ("one two three") for training data, and lorem ipsum text for validation. This approach maintains the problem-answer correspondence while removing the explicit reasoning trace, compelling the model to develop latent computational pathways. We fine-tune Qwen3-8B using identical hyperparameters to the encoded model: LoRA adaptation with rank $r = 1$ and scaling $\alpha = 256$ targeting all projection matrices, 4-bit quantization, and mixed precision training over 5 epochs (learning rate $2 \times 10^{-5}$, batch size 16 via gradient accumulation). The loss supervision on both filler tokens and final answers encourages the model to maintain answer accuracy despite the absence of meaningful intermediate steps. This training regime tests whether transformer architectures can learn to perform multi-step reasoning through internal representations when surface-level chain-of-thought is replaced with uninformative padding.

### C.3  POST HOC MODEL ORGANISM

To produce a model organism displaying post-hoc reasoning, we use similar SFT procedure as the above, except that we train the model on original CoT (same as baseline), with a prompt to ensure that the model already knows the answer prior to generating the CoT, producing reasoning that is inevitably post-hoc. Specifically, we append the answer to each question at the end of the prompt. To give an example: "Question: Given a binary string, return the minimum number of character swaps to make it alternating, or -1 if it is impossible.The string is called alternating if no two adjacent characters are equal. For example, the strings 010 and 1010 are alternating, while the string 0100 is not. Any two characters may be swapped, even if they are not adjacent. Now, determine the minimum number of swaps to make the following binary string alternating: 10001111001110. The CORRECT answer is -1. Write your reasoning in your reasoning steps...".

## D  ROBUSTNESS OF CoT PARAPHRASABILITY TO PARAPHRASING STYLE

### D.1  SENSITIVITY ANALYSIS ON GSM8K DATA

The paraphrasability metric in §3.4 depends on a paraphrasing operator applied to the sampled CoT. To check that our conclusions do not hinge on a particular paraphrasing template. We recompute

Table 3: **Sensitivity analysis for Paraphrasability metric.** We vary the paraphrasing mode (strength vs. length) and model (Gemini vs. GPT) to assess metric robustness. All configurations yield consistent negative values for the Encoded model organism.

| Configuration | Paraphraser | Mean | Std |
|---|---|---|---|
| Strength (original) | Gemini-2.0-flash | $-0.037$ | 0.059 |
| Strength | GPT-4o-mini | $-0.048$ | 0.058 |
| Length | Gemini-2.0-flash | $-0.020$ | 0.043 |

the CoT paraphrasability scores for Qwen3-8B on GSM8K under varying paraphrasing schemes: length-based rewrites, sentiment-targeted rewrites (positive and negative), and random-span paraphrasing. Each scheme uses the same set of paraphrase strengths as described in Appendix B.

Table 4 reports the distribution of paraphrasability scores for Qwen3-8B under each scheme. Across all four modes, scores concentrate in a similar range and show overlapping interquartile intervals, indicating that the aggregate behaviour of the metric is stable under changes to the paraphraser family.

Table 4: **CoT paraphrasability for Qwen3-8B across paraphrasing schemes.** Median and interquartile range (first quartile, third quartile) of the CoT paraphrasability metric over $n = 100$ GSM8K questions, for four paraphrasing modes applied to the sampled CoT.

| Paraphrasing mode | Median $(Q_1, Q_3)$ |
|---|---|
| `length` | $-0.176\,(-0.239, -0.053)$ |
| `positivity_strength` | $-0.103\,(-0.212, -0.007)$ |
| `negativity_strength` | $-0.093\,(-0.248, -0.010)$ |
| `section_random` | $-0.015\,(-0.157, 0.022)$ |

To assess robustness at the level of individual questions, we compute Spearman rank correlations between per-question paraphrasability scores obtained under different paraphrasing schemes (after aligning on the shared subset of questions for each pair). As shown in Table 5, scores are moderately to strongly correlated across schemes that paraphrase the entire CoT (`length`, `positivity_strength`, `negativity_strength`), with $\rho$ between $0.49$ and $0.66$. Correlations involving `section_random` are somewhat lower but remain positive. The results suggest that CoT paraphrasability reflects a stable per-question property of Qwen3-8B, rather than an artefact of any specific paraphrasing template.

Table 5: **Cross-mode Spearman correlations of CoT paraphrasability for Qwen3-8B.** Spearman rank correlation between per-question paraphrasability scores under different paraphrasing schemes on GSM8K.

| | `length` | `pos._strength` | `neg._strength` | `sect._random` |
|---|---|---|---|---|
| `length` | 1.00 | 0.49 | 0.63 | 0.28 |
| `pos._strength` | 0.49 | 1.00 | 0.66 | 0.27 |
| `neg._strength` | 0.63 | 0.66 | 1.00 | 0.48 |
| `sect._random` | 0.28 | 0.27 | 0.48 | 1.00 |

