# OpenReview forum: "House, G.P.T.: Diagnosing Pathological Chain-of-Thought in Reasoning Models"
_ICLR.cc/2026/Conference — Submitted to ICLR 2026_

### Official Review · Reviewer_CQyF · 2025-11-01

**Soundness:** 2
**Presentation:** 3
**Contribution:** 2
**Rating:** 4
**Confidence:** 4

**Summary:**

This paper introduces three task-agnostic health metrics (reliance, paraphrasability, and substitutability) to diagnose pathological chain-of-thought (CoT) reasoning in large language models. The authors create "model organisms" - models deliberately fine-tuned to exhibit specific CoT pathologies (encoded reasoning, internalized reasoning, and post-hoc reasoning) - and demonstrate that their metrics can successfully differentiate between these pathologies. They apply their diagnostic framework to several open-weight models, finding that most display relatively healthy CoT signatures on GSM8K.

**Strengths:**

The paper addresses an important problem for AI safety and model interpretability. The systematic approach to creating model organisms for each pathology type is well-structured, and the metrics are computationally inexpensive and model-agnostic, making them practical for deployment. The experimental methodology clearly demonstrates that each pathology produces distinct metric signatures, enabling differential diagnosis. The paper is generally well-written with clear explanations of the pathologies and metrics.

**Weaknesses:**

My main concern is that the model organisms are created through highly artificial training procedures, which may not reflect how these pathologies would naturally emerge in practice. For the encoded reasoning organism, training on a fixed animal-themed codebook is far removed from realistic steganographic encoding that might develop during training (such as in [1]). I also notice a lack of references to existing LLM steganography literature. Similarly, the internalized reasoning organism is trained explicitly on lorem ipsum text, which seems unlikely to occur naturally. The post-hoc organism simply appends answers to questions, which is definitional rather than emergent. This raises questions about whether the metrics would successfully detect naturally-occurring pathologies versus these synthetic versions.

Furthermore, the wild-type results are difficult to interpret. For instance, GPT-OSS-20B shows extremely high reliance scores (18.9), which the authors attribute to memorization of GSM8K answers. However, high reliance is supposed to be a sign of healthy CoT according to Table 1. This suggests the metrics may not be capturing what they intend to measure, or that the expected signatures in Table 1 are incomplete.

The metrics may also conflate training distribution effects with true pathologies. As the authors acknowledge, "interventions like removing CoT or substituting it with random text may push some model families further off-distribution than others due to differences in their training procedures" (Section 6). The paper does not provide a clear way to distinguish between low metric scores caused by genuine pathologies versus those caused by distribution shift, particularly for the substitutability metric.

Other relatively minor weaknesses:
- Despite using the word "causal interventions" (such as in Section 7 Conclusion), the metrics infer pathology from output statistics but do not verify internal causal mechanisms. This contrasts interpretability's normal definition of causal interventions.
- The paraphrasability metric relies on another LLM (Gemini-1.5-flash) to generate paraphrases, introducing an additional source of potential bias and error. Different paraphrasers might produce different results, and there is no analysis of how robust the metric is to the choice of paraphrasing model. Additionally, the paper mentions using "styles" (short, polite, negative, verbose, typos, reversal) but provides limited justification for why these particular styles were chosen or how they relate to detecting encoded reasoning.

---

[1] Motwani, Sumeet, et al. "Secret collusion among ai agents: Multi-agent deception via steganography." Advances in Neural Information Processing Systems 37 (2024): 73439-73486.

**Questions:**

The paper lacks comparison to existing faithfulness metrics from prior work [Lanham et al., 2023; Tutek et al., 2025]. How do these metrics perform relative to or in combination with existing approaches? Are they complementary or redundant? This comparison would strengthen the contribution.

The paper acknowledges a critical limitation but does not adequately address it: "Qwen3-8b should not be interpreted as a baseline for non-pathological CoT since various forms of unfaithfulness have been reported within reasoning models" (Section 6). If the baseline model used to validate the metrics may itself have pathologies, how can we trust that the metrics are correctly identifying pathological versus healthy CoT?

---

> ### Author Response · Authors · 2025-12-04
>
> We thank the reviewer for their thoughtful and detailed review.
>
> ### On artificial model organisms
>
> We agree that our model organisms are constructed rather than emergent. This is intentional—they provide *ground-truth validation* for our metrics. The revised paper's key finding addresses the concern about real-world applicability: pathological signatures are most pronounced at early training checkpoints (Figures 2-3). This suggests our metrics are most valuable as *early warning indicators* during training, before pathologies become subtle or entrenched. We cannot detect pathologies in fully-trained wild models if we cannot first validate detection on known cases.
>
> ### On GPT-OSS-20B's high reliance scores
>
> We have removed wild-type analysis from the revised paper to focus on the model organism validation. The reviewer correctly identifies that extreme logprobs from memorization confound interpretation—this reinforces our decision to use binary-alternation, which is less likely to be in training data.
>
> ### On distribution shift confounds
>
> This is a valid concern we acknowledge in Section 6. Our within-model comparison approach (comparing model organisms to the same architecture's pre-fine-tuning state) helps isolate pathology-inducing effects from architectural differences. The temporal analysis across checkpoints further helps—we observe *changes* in metrics as pathological training progresses, not just absolute differences.
>
> ### On "causal interventions"
>
> We use this term in the sense of Pearl's do-calculus: we intervene on the CoT (removing, paraphrasing, or substituting it) and observe effects on answer probability. We agree this differs from mechanistic interpretability's internal interventions and will clarify this distinction.
>
> ### On comparison to prior methods (Q1)
>
> We acknowledge we have not yet conducted direct comparisons to Lanham et al. (2023) or Tutek et al. (2025). This remains important future work. Our contribution differs in emphasis: we provide metrics with *distinct diagnostic signatures* for different pathology types, enabling differential diagnosis rather than binary detection.
>
> ### On baseline model validity (Q2)
>
> We define "healthy" operationally as the model's state before pathology-inducing fine-tuning. This relative baseline approach is explicit in our framing—we measure *changes* from baseline rather than claiming absolute health. The temporal analysis (tracking metrics across checkpoints) makes this relative approach concrete and practically useful.

---

### Official Review · Reviewer_vNma · 2025-11-01

**Soundness:** 2
**Presentation:** 1
**Contribution:** 2
**Rating:** 2
**Confidence:** 3

**Summary:**

This paper considers whether CoT explanations from large language models genuinely reflect their reasoning. It identifies three failure modes---post-hoc rationalization, encoded reasoning, and internalized reasoning---and proposes three corresponding diagnostic metrics: 1) reliance, 2) paraphrasability, and 3) substitutability. The metrics are validated on synthetically finetuned models ("model organisms") exhibiting these pathologies and applied to real open-weight models on GSM8K. The paper is timely and introduces a clear framework. However, its empirical scope is narrow (mostly GSM8K), interpretation of metric outputs can be unclear (large variance, unclear thresholds), and practical deployment challenges (e.g., intervention off-distribution effects, API constraints) are underexplored. Overall, it's an interesting contribution but would benefit from broader experiments without overclaiming and clearer guidance for practical use.

**Strengths:**

1. Addresses a timely and important problem: whether CoT is trustworthy for monitoring reasoning in LLMs.

2. Clear taxonomy of three CoT pathologies and corresponding diagnostic metrics (reliance, paraphrasability, substitutability).

3. Use of fine-tuned "pathological" models provides controlled validation of metrics.

4. Empirical results on real open-weight LLMs add practical relevance.

5. Writing is clear and limitations are explicitly acknowledged.

**Weaknesses:**

1. Overbranding / overstated novelty: The term “model organisms” is unnecessary rebranding of controlled/synthetic fine-tuning, and the paper overclaims originality.

2. Evaluation is very narrow---focused primarily on GSM8K math reasoning. unclear generalization to other reasoning tasks or domains.

3. Metric interpretations are unstable (high variance, unclear thresholds), making "healthy vs pathological" judgments ambiguous.

4. Practical deployment challenges (API-accessible models, off-distribution interventions, computational cost) are underexplored.

5. No ground truth for "pathology" in real models/claims of healthiness are inferred, not verified.

**Questions:**

1. Why is the term "model organism" preferable to simply "controlled fine-tuned model"? What conceptual or methodological value does the rebranding add?

2. How do the metrics perform outside GSM8K (e.g., commonsense reasoning, multi-hop QA, or open-ended tasks)?

3. Can you provide guidance or thresholds for interpreting metric values in practice? When is a model “pathological” vs “healthy”?

4. How sensitive are the metrics to paraphrase quality, model size, or prompt design?

5. For real-world deployment (especially black-box APIs), how feasible is it to compute these metrics without access to full log-probabilities or ability to modify CoT?

---

> ### Author Response · Authors · 2025-12-04
>
> We thank the reviewer for their detailed feedback.
>
> ### On "model organisms" terminology
>
> This term reflects established methodology in AI safety research (Hubinger et al., 2024). It is not "rebranding"—it describes a specific approach where we construct systems with known properties to validate measurement tools. The conceptual value is precision: "controlled fine-tuned model" does not convey that these are *deliberately constructed test cases for diagnostic validation*, analogous to how biologists use model organisms to study phenomena under controlled conditions.
>
> ### On evaluation scope
>
> We have moved from GSM8K to binary-alternation, a task requiring genuine reasoning (100% with CoT vs. 18% without). We acknowledge that broader evaluation across tasks remains important future work.
>
> ### On metric interpretation and thresholds
>
> Our metrics are designed for *relative comparison*, not absolute thresholds. The key insight from our revision is temporal: Figure 2 shows metric trajectories across training checkpoints, and Figure 3 shows Cohen's d effect sizes. Practitioners should monitor for significant shifts from baseline during training, using effect size conventions (|d| > 0.8 as large) as guides.
>
> ### On practical deployment (Q5)
>
> For API-only models without logprob access, our metrics would require approximate methods (e.g., sampling-based probability estimation). We acknowledge this limitation and note it explicitly in Section 6.
>
> ### On paraphrase sensitivity (Q4)
>
> Our sensitivity analysis (Table 3) shows consistent results across paraphrasing strategies (strength vs. length) and model families (Gemini-2.0-flash vs. GPT-4o-mini), demonstrating robustness to these choices.

---

### Official Review · Reviewer_GfiK · 2025-11-01

**Soundness:** 2
**Presentation:** 2
**Contribution:** 1
**Rating:** 2
**Confidence:** 4

**Summary:**

This paper presents an evaluation framework for detecting problematic CoTs in large reasoning models. The authors first characterize problematic CoTs as “pathologies” and divide them into three categories based on previous work: post-hoc rationalization, encoded reasoning, and internalized reasoning. They propose three metrics for evaluating these issues: reliance, paraphrasability, and substitutability. The measurement relies on comparing answer log probabilities before and after interventions on the CoT. To validate their metrics, they create “model organisms” by fine-tuning models to deliberately induce the pathological behaviors.

**Strengths:**

- The paper proposes new metrics to evaluate problematic behaviors in long CoTs from reasoning models.
- The proposed metrics are model-agnostic, task-agnostic, and computationally cheap (just log probability comparisons).
- The metrics capture different aspects of CoTs.

**Weaknesses:**

**1. Model organisms may not reflect pathologies.**
- I find the primary method of validating the metrics - fine-tuning capable models to exhibit some problematic behaviors - not convincing. It creates artificial cases that do not imply real unfaithful or pathological CoT cases in the wild. Real pathologies might emerge more subtly and be harder to detect.
- All wild-type models tested appear healthy (Table 3), so we don't know if metrics work on real pathologies.
- How well could the metrics detect pathologies "in the wild"?

**2. Interpretations of the metrics are unclear.**
- Is there a range (or bounds) for the metrics? Table 1 simply shows the metrics can be "high" or "low", but there is no way to quantify how high is too high.
- Why are both high and low values possible for some pathologies?
- Table 3: The use of log probs can make the metric values very extreme and less interpretable. Why are the values not normalized?

**3. The pathologies are all based on prior work, making the paper's contribution weak.**
- The described pathologies are all not new - CoT unfaithfulness is a longstanding problem that has been studied before reasoning models (https://aclanthology.org/2020.acl-main.386/, https://aclanthology.org/2023.ijcnlp-main.20.pdf).
- The main contribution of the paper seems to be merely characterizing existing CoT behaviors in a different way. The metrics are only describing behaviors that have already been shown, so I find there is a lack of insights from the proposed evaluation scheme.

**Questions:**

1. What does "House, G.P.T." mean in the title?

For other questions, please refer to Weaknesses section.

---

> ### Author Response · Authors · 2025-12-04
>
> We thank the reviewer for engaging with our work. We address each concern below.
>
> ### On "House, G.P.T." (Q1)
>
> This is a reference to *House, M.D.*, the medical drama where Dr. House diagnoses unusual pathologies through differential diagnosis. Our framework similarly performs differential diagnosis of CoT pathologies through distinct metric signatures.
>
> ### On model organisms not reflecting real pathologies
>
> We believe this criticism conflates two distinct goals. Our contribution is a *diagnostic toolkit*—metrics that can detect specific properties when present. Model organisms provide ground-truth validation that our metrics measure what they claim to measure. This is standard methodology: one validates a thermometer using known temperatures before trusting it in unknown conditions.
>
> Critically, our revised experiments reveal that pathological signatures are most pronounced at early training checkpoints and attenuate over time (Figure 2, Figure 3). This temporal dynamic suggests our metrics function best as *early warning indicators during training*—detecting emerging pathologies before they become entrenched. This reframes the practical application: developers can integrate these metrics into training loops for continuous monitoring.
>
> ### On metric interpretation
>
> We have added Cohen's d effect sizes (Figure 3) to provide standardized comparisons between healthy and pathological distributions. We frame our metrics as inherently comparative rather than absolute—the goal is to detect *relative shifts* from baseline behavior, particularly during training. Table 2 now provides expected effect size directions for each pathology type.
>
> ### On contribution relative to prior work
>
> While the pathologies themselves are known, our contribution is the first *operational diagnostic framework* with validated metrics that produce distinct signatures for each pathology type. Prior work demonstrated these phenomena exist; we provide tools to detect them at scale.

---

### Official Review · Reviewer_stHw · 2025-11-03

**Soundness:** 2
**Presentation:** 2
**Contribution:** 3
**Rating:** 4
**Confidence:** 4

**Summary:**

This paper studies several types of unfaithful reasoning, where the chain of though (CoT) of a reasoning language model (RLM) is misleading in some way. The three types studied are:
1. Post-hoc rationalization - The response is already determined and the RLM constructs a rationale rather than reasoning to reach an answer.
1. Encoded reasoning - The CoT is expressing some information, but in a way that the RLM understands and monitors do not.
1. Internalized reasoning - Some or all of the true reasons for an RLM's response are not expressed in the CoT.

The paper proposes several metrics to test for these types of unfaithful reasoning. Since understanding the true relationship between a CoT and a final response is very difficult, the paper proposes to validate the metrics with RLMs that have been specifically fine-tuned to exhibit these behaviors. Experiments show that the metrics are accurate at recovering known instances of unfaithful reasoning. The paper then applies these metrics to popular RLMs and argues that they are mostly faithful in their reasoning.

**Strengths:**

1. The paper takes an interesting approach to studying a very important problem. The idea of creating deliberately unfaithful RLMs to validate metrics is novel to my knowledge.

2. The metrics are straightforward and easy to implement.

3. The metrics could see wide use as validation tools in the development and application of RLMs.

**Weaknesses:**

1. I really do not care for dressing up the paper in the language of biology and medicine. I think referring to types of unfaithful reasoning as "pathologies" and fine-tuned RLMs as "model organisms" does not add anything to the paper. It only risks exaggerating hype around AI.

2. As far as I can tell, all experiments were done on GSM8k. (Section 4.1 called "Models and Datasets" does not mention any datasets, so it is unclear.) This dataset seems likely to be similar to the training data for many RLMS. (In fact Section 4.3 guesses that some models have memorized this dataset.) It seems premature to declare these models mostly "healthy" if they haven't been evaluated on anything more challenging or novel to the models.

**Questions:**

1. How different are the abilities between the "model organisms" and "wild type" models? Are the model organisms as accurate? Perhaps they are either not that realistic as models or perhaps poor performance would also reveal these "pathologies?"

2. The text in Figure 3 is too small to read.

---

> ### Author Response · Authors · 2025-12-04
>
> We thank the reviewer for their thoughtful feedback and recognition that our metrics "could see wide use as validation tools."
>
> ### On terminology ("pathologies" and "model organisms")
>
> We respectfully disagree that this terminology "exaggerates hype around AI." The term "model organisms" is established in the AI safety literature (Hubinger et al., 2024, "Model Organisms of Misalignment") and serves a precise methodological purpose: just as biologists use *Drosophila* to study genetics in controlled conditions before investigating wild populations, we deliberately construct models with known properties to validate diagnostic tools before applying them in the wild. This is not mere rebranding—it reflects a specific research methodology where ground-truth labels are available by construction. We believe this framing clarifies rather than obscures our contribution.
>
> ### On dataset scope
>
> We appreciate this concern. In our revision, we have moved from GSM8K to the binary-alternation dataset from reasoning-gym, which offers several advantages: (1) it requires genuine reasoning (100% accuracy with CoT vs. 18% without), (2) it is less likely to be memorized by foundation models, and (3) the reasoning path is verifiable. We acknowledge that expanding to additional datasets remains valuable future work.
>
> ### On model organism accuracy (Q1)
>
> Our model organisms maintain comparable accuracy to wild-type models on tasks where they can apply their pathological reasoning strategies. This is by design—a pathological model that simply fails would not be a useful test case. The key difference is *how* they arrive at answers, which our metrics are designed to detect.
>
> ### On Figure 3
>
> We have revised this figure to improve legibility. Thank you for flagging this.

---

### Meta-Review · Area_Chair_RNm5 · 2026-01-06

**Summary:**

The paper studies common issues of CoT reasoning such as post-hoc rationalization of a certain answer and worsening interpretability of CoT, and proposes a set of "health metrics" to assign a binary score healthy/unhealthy relative to synthetically fine-tuned baselines exhibiting a given behavior.

Reviewers are rather skeptical about the work raising concerns about:
* a very narrow scope of the experiments (stHw, vNma) on GSM8k only (later replaced with another hand-crafted problem),
* problems of interpretability of yes/no scores and causal interpretation of the scores (GfiK, vNma),
* synthetic nature of the "model organisms" which are unlikely to happen in real-world LLMs (GfiK, CQyF),
* questionable novelty (stHw, vNma) by re-packaging well-known issues with a "medical" language - and I agree it does a great deal of disservice to the paper
* the eval protocol can't be applied to models hidden behind APIs (vNma)

The rebuttal stage (more details below) did not bring sufficient clarity and experimental evidence, and I believe the paper is not yet ready for ICLR, so I recommend a reject.

**Reviewer Concerns:**

The concerns remain largely unaddressed.

* Narrow experimental scope: GSM8k got replaced by another hand-crafted dataset (but there is an abundance of reasoning benchmarks) which is rather insufficient
* Interpretability: the authors removed a seemingly contradictory finding about GPT-OSS but explanations/guidance on how to add a binary yes/no score instead of a numerical value remain quite vague
* A synthetic nature of the probes is motivated by the necessity of "early warning indicators" during training, but the experiments entirely omit RL tuning (which might be responsible for some of the symptoms) and only track SFT stage of a 8B model.
* Novelty and the eval protocol remain open issues.

**Reviewer Scores:**

I don't think reviewers would have changed their scores.

---

### Decision · Program_Chairs · 2026-01-26

Reject